# Temporal Profiling of the Cortical Synaptic Mitochondrial Proteome Identifies Ageing Associated Regulators of Stability

**DOI:** 10.3390/cells10123403

**Published:** 2021-12-02

**Authors:** Laura C. Graham, Rachel A. Kline, Douglas J. Lamont, Thomas H. Gillingwater, Neil A. Mabbott, Paul A. Skehel, Thomas M. Wishart

**Affiliations:** 1The Roslin Institute, University of Edinburgh, Easter Bush Campus, Midlothian EH25 9RG, UK; laura.graham@roslin.ed.ac.uk (L.C.G.); rachel.kline@roslin.ed.ac.uk (R.A.K.); neil.mabbott@roslin.ed.ac.uk (N.A.M.); 2Euan MacDonald Centre, Chancellor’s Building, University of Edinburgh, 49 Little France Crescent, Edinburgh EH16 4SB, UK; T.Gillingwater@ed.ac.uk (T.H.G.); Paul.skehel@ed.ac.uk (P.A.S.); 3FingerPrints Proteomic Facility, College of Life Sciences, University of Dundee, Dow Street DD1 5EH, UK; d.j.lamont@dundee.ac.uk; 4Centre for Discovery Brain Sciences, Hugh Robson Building, George Square, Edinburgh EH8 9XD, UK; 5Centre for Dementia Prevention, The University of Edinburgh, 9A Bioquarter, 9 Little France Road, Edinburgh EH16 4UX, UK

**Keywords:** mitochondria, synapse, aging, proteomics, neuron

## Abstract

Synapses are particularly susceptible to the effects of advancing age, and mitochondria have long been implicated as organelles contributing to this compartmental vulnerability. Despite this, the mitochondrial molecular cascades promoting age-dependent synaptic demise remain to be elucidated. Here, we sought to examine how the synaptic mitochondrial proteome (including strongly mitochondrial associated proteins) was dynamically and temporally regulated throughout ageing to determine whether alterations in the expression of individual candidates can influence synaptic stability/morphology. Proteomic profiling of wild-type mouse cortical synaptic and non-synaptic mitochondria across the lifespan revealed significant age-dependent heterogeneity between mitochondrial subpopulations, with aged organelles exhibiting unique protein expression profiles. Recapitulation of aged synaptic mitochondrial protein expression at the *Drosophila* neuromuscular junction has the propensity to perturb the synaptic architecture, demonstrating that temporal regulation of the mitochondrial proteome may directly modulate the stability of the synapse in vivo.

## 1. Introduction

Mitochondria are highly dynamic and heterogeneous organelles that have the propensity to directly modulate synaptic architecture. Although mitochondria have long been associated with the dysfunction of neuronal compartments during advancing age [1], the mechanisms orchestrating such events and how these may impact cognitive function remain poorly understood. Traditional theories of age-dependent mitochondrial dysfunction, such as the free radical theory of ageing [2], which suggests that mitochondrial reactive species cumulatively promote cellular damage during the lifespan—are now beginning to be revised due to novel experimental evidence challenging numerous facets of the model [1,2,3,4]. In recent years, a wealth of innovative studies across numerous animal models have promoted the development of an alternative multi-factorial model of mitochondrial ageing, suggesting to varying degrees that alterations in reactive species detoxification [5,6,7,8,9], bioenergetics [10,11,12,13], Ca^2+^ buffering capacity [14,15], mtDNA integrity [16,17,18,19], and organelle dynamics [20,21] synergistically facilitate changes in neuronal receptivity during advancing age.

Synapses are particularly susceptible to the effects of advancing age; however, the biochemical cascades governing this compartmental vulnerability remain elusive. Accordingly, recent investigations have begun to delineate how temporal alterations in mitochondrial subpopulations may be contributing to perturbations in the synaptic milieu and facilitating concomitant cognitive decline [22]. Indeed, morphometric analyses of mitochondria using 3D electron microscopy techniques in young and aged rhesus macaques revealed significant age-dependent morphological alterations in organelles within cortical presynaptic terminals, which robustly correlates with cognitive capacity [20,21]. Young animals exhibited stereotypical spherical and tubular mitochondria, whereas old animals appeared to harbour numerous “donut” shaped organelles, which are considered to be indicative of mitochondrial dysfunction. Intriguingly, the number of “donut” mitochondria directly correlated with the size of the presynaptic active zone and a number of synaptic vesicles, in addition to the severity of cognitive impairment, suggesting that divergent mitochondrial morphologies may modulate synaptic transmission, plasticity and stability during advancing age [20,21,23]. More recently, three-dimensional electron microscopy mapping of neuronal mitochondria revealed cell compartment-specific morphological alterations detectable through different subregions of the ageing murine hippocampus, suggesting that both neuroanatomical and compartmentalised selectivity in mitochondrial dynamics accompanies ageing-associated cognitive decline [22]. In agreement with this, investigations examining synaptic mitochondrial motility have reported that the presence of mitochondria within the presynaptic compartment dictates the strength of transmission [24,25]. Retrograde trafficking of mitochondria away from the presynaptic terminal mediates reductions in synaptic strength, whereas the presence of the organelles in apposition to the active zone promotes sustained neurotransmitter release and plastic modifications within the synaptic terminal [24,25], these data indicate that synaptic mitochondria play a fundamental role in regulating a multitude of properties at the synapse [23] and dynamic temporal modifications of this discrete population of organelles may significantly impact upon the structure and function of the compartment, promoting alterations in cognitive capacity.

The outlined studies documenting the dynamic intersection between mitochondrial morphology and synaptic function during ageing are particularly compelling, highlighting the influence of mitochondria on cognition. However, molecular investigations attempting to unravel the causal and/or concomitant temporal biochemical alterations regulating this shift in synaptic mitochondrial morphometry demonstrate significantly variable results, with reports describing conflicting changes in important modulatory processes such as respiratory capacity [10,12,13,26,27,28] and the fission/fusion balance [27,28,29]. Indeed, a proteomic study by Stauch et al., examining the temporal alterations occurring in the mouse synaptic mitochondrial proteome, reported that increased age promoted the progressive downregulation of mitochondrial electron transport chain subunits; however, this appeared to have no functional impact on the bioenergetic capacity of isolated synaptic mitochondria [27]. Despite this, there are now a number of studies disputing these results and investigations employing similar biochemical methodologies reveal that synaptic mitochondrial respiratory capacity appears to be significantly diminished with advancing age [26,30]. Although the adoption of proteomic methodologies to track the molecular spatiotemporal alterations occurring in synaptic mitochondria is a valid approach, a non-synaptic mitochondrial population for proteomic comparison was omitted, preventing delineation of how subpopulation-specific biochemical alterations may be modulating compartmental vulnerability during ageing.

Mitochondria exhibit subcellular heterogeneity in morphology and protein expression [31], and studies are now beginning to demonstrate that the synaptic mitochondrial proteome may be dynamically altered during ageing [27,28,32]. However, the presence of discrepancies regarding the temporal regulation of mitochondrial properties in the literature has hampered the identification of functional mitochondrial mediators that may be modulating synaptic stability and cognitive decline during advancing age. There is significant demand for well-executed, physiologically relevant studies that characterise the spatiotemporal molecular alterations occurring in both synaptic and non-synaptic mitochondria to enable the identification of biochemical cascades that may be promoting age-dependent vulnerability of the synaptic compartment.

In order to address this, here we employ an unbiased combinatorial approach, including quantitative proteomics coupled with in vivo phenotypic assessments, to temporally profile the biochemical alterations occurring in mitochondrial subpopulations throughout normal healthy ageing in the rodent cortex. We demonstrate that mitochondrial ageing is highly heterogeneous, and subcellular localisation dictates the biochemical composition of discrete mitochondrial populations throughout the lifespan. Strikingly, we reveal that over 1000 proteins are temporally altered by greater than 20% in expression in both synaptic and non-synaptic mitochondrial populations across the time course, demonstrating significant age-dependent modifications at the proteome level. Interestingly, aged synaptic mitochondria exhibit considerable divergence in global protein expression, which may contribute to the enhanced vulnerability of cortical synapses at this particular age. Recapitulation of aged synaptic mitochondrial protein expression using molecular genetic tools in vivo revealed several novel functional mediators that have the propensity to significantly modulate multiple morphological parameters at the synapse. We suggest that selective alterations in the synaptic mitochondrial proteome may contribute to the documented structural and functional perturbations occurring at synaptic terminals during advancing age.

## 2. Methods

### 2.1. Ethics

In compliance with the 3Rs, no animals were bred specifically for this project. All tissue samples used in the current study were derived from existing archived brains or harvested from wild-type colonies maintained at the Roslin Institute. All animal experiments were approved by The University of Edinburgh internal ethics committee. As no animals were bred specifically for this study, our investigation was designed to characterise the molecular (rather than functional) alterations detectable during the neuronal ageing process in rodents.

### 2.2. Animals

Twelve wild-type C57BL/6 mice of different sexes and ages were utilised for the studies described. Animals were assigned to the following groups: young (mean age = 4 weeks), mid-age (mean age = 6 months), or old age (mean age = 24 months) group, each consisting of *n =* 4 animals per time point. Animals were terminated by an overdose of CO_2_, and their brains were immediately excised. Brain stem and cerebellum were removed and discarded, and the remaining forebrains were frozen slowly in an insulated container at −80 °C to preserve morphological integrity. Immediately prior to mitochondrial isolations, forebrains were rapidly thawed by immersion in a water bath at 37 °C (for further information on slow-freeze/rapid thaw methodology, please refer to Hardy et al., 1983 [33]. Samples were weighed and pooled by age group for homogenisation and mitochondrial preparations, thus generating three technical replicates per age group, similar to methodologies described in [31,32].

### 2.3. Mitochondrial Isolations

For a comprehensive methodological outline, see Lai et al. 1977 [34]. This method has been published as yielding a high level of enrichment for mitochondrial proteins and, more recently, as sensitivity has increased an increasing number of strongly associated proteins and conserves biochemical activity [31]. For full details, please see [31] and the accompanying online data repository (https://doi.org/10.7488/ds/3181, accessed on 15 November 2021).

### 2.4. Label-Free Proteomics

Synaptic and non-synaptic mitochondrial preparations were extracted in SDT lysis buffer containing 100 mM Tris-HCl, 4% (*w*/*v*) Sodium dodecyl sulfate (VWR) and 0.1 M d/l-dithiothreitol (Sigma) (pH 7.6). For efficient protein extraction, lysates were freeze-thawed and homogenised in SDT buffer several times. Protein concentration was then determined using a BCA assay. Aliquots (200 μg) of each preparation were processed through FASP (filter-aided sample preparation) involving buffer exchange to 8 M urea and alkylation with 50 mM iodoacetamide prior to double digestion with trypsin (Roche, sequencing grade), initially for 4 h (at 37 °C), then overnight at 30 °C. Trypsin-digested peptides were separated using an Ultimate 3000 RSLC (Thermo Scientific, San Jose, CA, USA) nanoflow LC system with the column oven set to 35 °C. Technical replicates (3 × ~1 μg) of each sample were loaded at a constant flow of 5 μL/min onto a trapping cartridge (PepMap100, C18, 5μm, 100 Å 0.3 × 5 mm; (Thermo Scientific, San Jose, CA, USA) using 2% Acetonitrile, 0.1% formic acid. After trap enrichment, peptides were separated on a peptide CSH, 1.7 μm, 130 Å, 75μm × 250 mm C18 column (Waters Corp, Milford, MA, USA) with the following gradient: t = 0 min, 2% B; t = 6, 2% B; t = 20, 8% B; t = 110, 24% B; t = 135, 37% B where solvent A is water with 0.1% formic acid and solvent B is 80% acetonitrile with 0.1% formic acid, with a constant flow of 260 nL/min. The HPLC system was coupled to a linear ion trap Orbitrap hybrid mass spectrometer (LTQ-Orbitrap Velos Pro, Thermo Scientific) via a nanoelectrospray ion source (Thermo Scientific). The spray voltage was set to 2.2 kV, and the temperature of the heated capillary was set to 200 °C. Full-scan MS survey spectra (*m*/*z* 335–1800) in profile mode were acquired in the Orbitrap with a resolution of 60,000 after accumulation of 1,000,000 ions. A lock mass of 445.120 024 was enabled for survey scans to improve mass accuracy. The 15 most intense peptide ions from the preview scan in the Orbitrap were fragmented by collision-induced dissociation (normalised collision energy, 35%; activation *Q*, 0.250; and activation time, 10 ms) in the LTQ after the accumulation of 10,000 ions. Dynamic exclusion parameters were set as follows: repeat count, 1; repeat duration, 30 s; exclusion list size, 500; exclusion duration, 45 s; exclusion mass width, plus/minus 10 ppm (relative to reference mass). Maximal filling times were 10 ms for the full scans and 100 ms for the MS/MS scans. Precursor ion charge state screening was enabled, and all unassigned charge states, as well as singly charged species were rejected. Data were acquired using Xcalibur software.

Raw proteomic data were imported into Progenesis for characterisation and analysis of relative ion abundance. Prior to alignment peptide ion filtering was performed to eliminate those with power < 0.8 and *p* > 0.05 (giving 825,600 Ms/Ms Spectra). 2D representations of MS/MS output were created for each sample, and these were aligned to determine similar features (average alignment score >80%). Following alignment, data were filtered by retention time with features detected below 12.16 min and above 133.78 min discarded to correct for elution variability. The runs were grouped according to age, and mitochondrial subcellular localisation and statistical *p* values were automatically generated in Progenesis software through a one-way ANOVA on the ArcSinh transform of the normalised data.

Peptides were filtered by the following criteria: power < 0.8, fold change > 1.2, *p* > 0.05 and the remaining data were exported from Progenesis for identification of individual peptide sequences using the IPI-*Mus musculus* database via Mascot Search Engine (V2.3.2). Enzyme specificity was set to that of trypsin, allowing for cleavage N-terminal to proline residues and between aspartic acid and proline residues. Other parameters used were as follows. (i) Variable modifications: methionine oxidation, methionine dioxidation, protein N-acetylation, gln → pyro-glu. (ii) Fixed modifications: cysteine carbamidomethylation. (iii) MS/MS tolerance: FTMS—10 ppm, ITMS—0.6 Da. (iv) Minimum peptide length: 6. (v) Maximum missed cleavages: 2. (vi) False discovery rate: 1%. A cutoff score of >34 was used based on Mascot probability threshold of 0.05 that an observed hit is a random event. As an indication of identification certainty, the false discovery rate for peptide matches above the identity threshold was set at 1% (as previously described, see Graham et al. 2017 [17]).

Identified proteins were re-imported into Progenesis for further processing. Proteins were subject to stringent filtering parameters to eliminate those which had <2 unique peptides and *p* > 0.05 to obtain the proteins which demonstrated statistically significant alterations in mitochondrial protein expression over the ageing time course. Progenesis outputs are freely available here: https://doi.org/10.7488/ds/3181, accessed on 15 November 2021.

### 2.5. BioLayout Express^3D^

Proteomic data were dissected using the complex pattern recognition software BioLayout Express^3D^ [35]. The software enables unbiased visualisation of molecular networks exhibiting similar expression profiles through the application of Markov clustering algorithms to raw proteomic data (MCL 2.2). All graphs were clustered using Pearson correlation r = 0.96. Clusters of interest indicating age-dependent alterations included those that demonstrated a steady up- or down-regulation or a late-stage up- or late-stage down-regulation during the time course of ageing. Proteins from clusters with analogous expression profiles underwent a subtractive process, as candidates appearing in both the synaptic and non-synaptic clusters, altered in the same manner, were unlikely to regulate synaptic stability during ageing and were therefore eliminated from further analyses.

### 2.6. Ingenuity Pathway Analysis

IPA analyses were performed as previously described [36,37] with the interaction data limited as follows: direct and indirect interactions; experimentally observed data only; 35 molecules per network; 10 networks per dataset. Prediction activation scores (z-score) were calculated in IPA. The z-score is derived from Fisher’s exact test and is a statistical measure of the match between an expected relationship direction as predicted by the Ingenuity Knowledge Database and the observed protein expression as detected in the input data. Positive z-scores indicate predicted activation of cascade (orange), while negative z-scores indicate predicted inhibition (blue) [37].

### 2.7. Drosophila Stocks

All stocks lines used for experiments were obtained from the BDSC with the following stock numbers: 9837, 26,650, and 42,580. Flies were raised on standard cornmeal food at room temperature. Homology of mouse gene of interest and respective *Drosophila* ortholog was determined by input into DIOPT (DRSC Integrative Ortholog Prediction Tool [38] see Table 1). The *elav-Gal4^C155^* driver strain was used for all experiments.

### 2.8. Immunohistochemistry

Wandering third instar larva were selected and dissected in PBS (*n* = 8). The dissected larval neuromuscular junctions (NMJs) were fixed in Bouin’s fixative (15:5:1 picric acid, 37% formaldehyde and acetic acid) for 10 min and washed thoroughly in PBT (PBS + 0.1% TritonX-100). Preparations were blocked in PBT + 10% normal goat serum for 2 h then incubated in primary antibody overnight at 4 °C. NMJs were again washed extensively in PBT and incubated in secondary antibody at room temperature for 2 h. Samples were mounted on microscope slides using Vectashield mounting medium (Vector Laboratories) and imaged on a Zeiss confocal microscope. All quantification was performed in ImageJ using a standardised protocol for NMJ analyses [39]. Primary antibodies: horseradish peroxidase (HRP), Jackson ImmunoResearch (323-005-021); bruchpilot (BRP/nc82), Developmental Studies Hybridoma Bank (nc82-s). Secondary antibodies: Alexa Fluor 488-AffiniPure Goat Anti-Rabbit IgG, Jackson ImmunoResearch (111-545-003); Cy3-AffiniPure Goat Anti-Mouse IgG, Jackson ImmunoResearch (115-165-146).

### 2.9. Statistical Analyses

Data were collected in Microsoft Excel, and statistical tests were performed in GraphPad Prism 6 software. For all analyses, *p* < 0.05 was considered significant. Statistical tests used are detailed in the results or figure legends where appropriate.

## 3. Results

### 3.1. Spatiotemporal Characterisation of Discrete Mitochondrial Proteomes

To identify age-dependent molecular alterations occurring in discrete neuronal compartments of the rodent brain, we purified and characterised isolated mitochondria from cortical synaptic and non-synaptic compartments at three distinct time points (young adult, mid-age, and old (Figure 1)). Quantitative label-free proteomic analyses identified >1800 common proteins common to each subpopulation across the timecourse (Figure 2A,B), revealing dynamic variations in synaptic protein expression. Markedly, over 1000 proteins were altered by greater than 20% in each discrete mitochondrial population, demonstrating significant age-dependent modifications at the proteome level (Figure 2C). To assess the purity of mitochondrial preparations, PANTHER Gene Ontology bioinformatics software was employed. The input of the 1857 identified proteins from the proteomic data indicated enrichment of the ATP synthase complex and the mitochondrial inner membrane suggesting relative purity of the preparations. For examination of synaptic enrichment in the synaptic mitochondrial fractions, we utilised the raw proteomic data to perform quantitative analyses. The normalised average abundances of the well-established synaptic markers synaptophysin, synaptic vesicle glycoprotein 2A (SV2A) and synaptotagmin-7 were calculated for both the synaptic and non-synaptic mitochondrial samples for comparison. Synaptophysin, SV2A and synaptotagmin-7 indicated significant enrichment in synaptic mitochondrial samples versus corresponding non-synaptic mitochondrial fractions suggesting purification of mitochondria derived from the synaptic compartment (Figure 2D).

### 3.2. Mitochondrial Ageing Demonstrates Subcellular Heterogeneity

To address whether dynamic variations in the mitochondrial proteome may be contributing to age-dependent synaptic vulnerability, we sought to determine if organelles from discrete subcellular compartments aged in a similar manner. Using the objective network visualisation software BioLayout Express^3D^ ([35] www.biolayout.org, accessed on 15 November 2021), we generated synaptic and non-synaptic mitochondria unbiased sample-sample correlation graphs demonstrating relative age-dependent similarities (Figure 3A,B). Examination of these networks confirmed distinct compartment-dependent clustering profiles. Non-synaptic mitochondria displayed a single network suggesting a degree of homogeneity between young, mid-age and old samples (Figure 3B). Though the old and young mitochondria appear to demonstrate reduced equivalence—as indicated by the distance between population nodes—the data suggest that there are fewer significant temporal alterations occurring in the non-synaptic versus the synaptic organelles. Conversely, the synaptic mitochondria demonstrated increased heterogeneity between sample populations. Fragmentation of the graph indicates that the aged synaptic mitochondrial population display discrete protein expression profiles (Figure 3A), which correlate with and may contribute to the enhanced vulnerability of cortical synapses at this particular age. Thus, there were indications that isolated mitochondrial populations derived from discrete subcellular compartments age heterogeneously and the unique underlying alterations occurring in synaptic mitochondria may dictate the potential vulnerability of synapses.

### 3.3. Temporal Protein Profiling of Discrete Mitochondrial Populations Reveals Protein Expression Trends Correlating with Synaptic Vulnerability

In conjunction with the unbiased sample-sample correlation analyses, to examine compartment-specific temporal mitochondrial alterations further, we generated network graphs of the synaptic and non-synaptic mitochondrial timecourse proteomic data, again utilising BioLayout Express^3D^ [35]. The software applies unbiased Markov clustering algorithms to the input data and groups proteins displaying similar expression trends, allowing visualisation of spatiotemporal profiles and identification of discrete biochemical cascades altered within the dataset.

Synaptic and non-synaptic mitochondrial graphs were constructed utilising the 1857 commonly identified proteins through the timecourse of ageing (see Figure 2A,B), providing 35–45 clusters per subpopulation.

By utilising two mitochondrial subpopulations demonstrating disparate profiles of ageing, we aimed to identify potential mitochondrial regulators of synaptic vulnerability with a comparison of analogous protein expression profiles. Clusters displaying particular expression trends of interest were selected (i.e., increasing or decreasing with age) from both the synaptic and non-synaptic mitochondrial timecourse network graphs generated in Biolayout Express^3D^ (Figure 4A,B). Those clusters exhibiting steady up- or down-regulation protein expression profiles during the timecourse were considered as correlates of normal healthy ageing due to the predictable age-dependent tractability of those candidates (Figure 5A). Conversely, proteins displaying late-stage increases or decreases in expression were regarded as correlates of synaptic and/or organelle vulnerability as the abrupt expression changes observed in the old synaptic and non-synaptic populations likely reflected acute alterations disrupting mitochondrial homeostasis (Figure 5B). In order to identify mitochondrial candidates that may be regulating cortical synaptic vulnerability during ageing, proteins from synaptic and non-synaptic mitochondrial clusters with analogous expression profiles were subject to a subtractive process. Proteins demonstrating equivalent spatiotemporal profiles in both synaptic and non-synaptic mitochondrial populations were not considered to be regulators of cortical synaptic vulnerability and filtered from the data prior to further analysis (Figure 5C,D). Following this subtraction, we considered the remaining 451 proteins, each exhibiting expression profiles harbouring a sudden “peak” or “trough” in old age, to be associated with age-dependent synaptic vulnerability (Figure 5A,B).

### 3.4. Temporal Regulation of the Synaptic Mitochondrial Proteome Can Modulate Synaptic Morphology

Despite characterising the global spatiotemporal changes occurring in discrete mitochondrial subpopulations, it remained unclear whether divergence in the expression of individual mitochondrial candidate proteins may be capable of actively regulating synaptic vulnerability. To elucidate mitochondrial regulators of synaptic stability, we initially mapped the individual temporal expression profiles of the 451 candidates identified by the BioLayout Express^3D^ analyses using Python Jupyter Notebook. Upon examination of the protein expression trends, we hypothesised that candidates likely capable of modulating synaptic stability would exhibit unequivocal temporal profiles across mitochondrial subpopulations, with considerable divergence at old age. Thus, proteins with the corresponding expression at the young and mid-age time-points in both synaptic and non-synaptic mitochondria, followed by a significant demarcation in expression at old age, were selected as potential regulators of synaptic vulnerability. These particular expression profiles correlate with previous reports of significant age-dependent alterations in mitochondrial morphology [40], bioenergetics [10,11], and calcium buffering capacities [41], which have the propensity to perturb synaptic structure and function [42].

Following the identification of 243 proteins demonstrating the archetypal spatiotemporal expression profile (Figure 6A,B), we further refined the candidate compendium by selecting proteins that exhibited a >2 fold-change in expression between synaptic and non-synaptic mitochondria at the old-age time point (Figure 6B). The remaining 96 candidates represented potential mitochondrial regulators of age-dependent synaptic vulnerability. Interestingly, DAVID functional annotation analyses of these mitochondrial candidates indicated that alterations in DNA methylation cascades might be contributing to synaptic vulnerability during advancing age (Figure 6C). To examine whether alterations in candidate protein expression in aged synaptic mitochondria was contributing to the vulnerability status of synaptic compartments in vivo, we utilised a molecular genetic approach at the *Drosophila* larval neuromuscular junction (NMJ) to assess the regulatory role of the individual candidates Rab31, RhoG, and Mcu at the synapse.

Recapitulation of candidate protein expression at the *Drosophila* larval NMJ was achieved using the *UAS*/Gal4 system, promoting pan-neuronal expression of the selected transgenes under the control of the *elav-Gal4* driver (see Table 1 for *Drosophila* orthologs). Regulated expression of the constructs (BDSC stock numbers: 9837, 26650, 42580) resulted in viable offspring from all crosses, allowing examination of individual candidate modulatory effects on the synaptic architecture. Manipulation of Rab31 (9837), RhoG (26650), and Mcu (42580) expression revealed striking synaptic phenotypes at the third instar larva muscle 12/13 NMJ in multiple morphological parameters associated with the structural and functional stability of the synapse, including altered patterning in conjunction with a quantifiable reduction in both total bouton and active zone punctate areas (Figure 7). Indeed, enhanced constitutive expression of the mitochondrial associated vesicular trafficking protein Rab31 promoted modest reductions in the total bouton area (*p* < 0.05) in addition to concomitant decreases in the active zone punctate size (*p* < 0.0001) versus the control line (Figure 7A–C). Although Rab31 NMJ arborisation appears to demonstrate considerable similarity to that of the control, minor reductions in the branching of distal processes may be contributing to the observed reductions in the total bouton area. Conversely, RhoG overexpression promotes marked perturbations in arborisation, with NMJs displaying truncated branches and aberrant compacted morphologies (Figure 7A). Correspondingly, significant reductions in bouton volume (*p* < 0.0001) and active zone punctate size (*p* < 0.0001) are observed (Figure 7A–C), which are likely mediated by the role of RhoG in the polymerisation of the actin cytoskeleton and regulation of dendritic differentiation and stabilisation during development [43]. RNAi-mediated knockdown of Mcu expression produced a morphologically distinct but quantitatively similar phenotype to that of RhoG manipulation (Figure 7A). Reductions in Mcu activation also appear to attenuate the stereotypic arborisation patterns of the NMJ, in addition to facilitating decreases in the total bouton area (*p* < 0.0001) and active zone punctate size ((*p* < 0.0001) Figure 7A–C). Despite all candidate lines (Rab31, RhoG, and Mcu) demonstrating a significant reduction in active zone punctate size, with quantification of total active zone staining per NMJ, we report no significant difference versus control lines. Although vesicular dynamics may be regulated via biochemical cascades independent of those manipulated here, the significant reduction in total bouton area in multiple candidate lines suggests that coordinated expression of these particular mitochondrial proteins may be essential for the continued maintenance of synaptic-specific structures during advancing age.

## 4. Discussion

How the spatiotemporal regulation of the mitochondrial proteome and synaptic function intersect during advancing age is poorly understood. Here, we indicate that selective biochemical alterations in the synaptic mitochondrial proteome may promote age-related perturbations in synapse structure and function in vivo. Temporal proteomic profiling of distinct subcellular mitochondrial populations from the rodent cortex revealed discrete and dynamic alterations in both the synaptic and non-synaptic mitochondrial proteomes during “normal healthy” ageing. Intriguingly, aged synaptic mitochondria appeared to develop a unique proteomic fingerprint. Recapitulation of aged synaptic mitochondrial protein expression using molecular genetic tools in vivo revealed several novel functional mediators that have the propensity to significantly modulate multiple morphological parameters at the synapse, suggesting that the mitochondrial proteome and synaptic morphometry are intimately intertwined. The data indicate that selective alterations in synaptic mitochondrial protein expression may, in part, mediate enhanced vulnerability of the cortical synapse during advancing age.

### Mitochondrial Candidates Modulating Synaptic Morphology

The identification of 96 candidates demonstrating a >2 fold change between synaptic and non-synaptic mitochondria in a temporal profile associated with increased vulnerability of the synapse is highly indicative of the regulatory role the organelles may play in modulating synaptic structure and function during ageing. Interestingly, the 96 differentially expressed candidates appeared to demonstrate enrichment in DNA methylation cascades. Recent evidence has suggested that the methylation of mtDNA is age- and brain region-dependent [44,45,46], and reductions in mtDNA methylation promote transcriptional changes associated with senescent phenotypes in vitro [47,48]. Furthermore, decreases in mtDNA methylation have also been associated with genomic instability, promoting mutagenesis and dysregulation of the respiratory complex genes encoded by the mitochondrion [45]. Strikingly, the data obtained in this study indicate that mtDNA methylation may also demonstrate discrete subcellular patterns, which suggests differential regulation of mtDNA integrity in distinct mitochondrial subpopulations. Although global perturbations in mtDNA stability have been widely associated with advancing age, few studies have examined whether mtDNA integrity differs in ageing synaptic and non-synaptic mitochondria. Our results suggest that this may warrant exploration to determine if synaptic mitochondria exhibit increased susceptibility to mtDNA mutagenesis, organelle dysfunction and concomitant compartmental instability during ageing.

Although we identified 96 candidates representing potential mitochondrial regulators of age-dependent synaptic vulnerability, as a proof of principle, we phenotypically assessed three candidates in vivo. Critically, we demonstrate that recapitulation of Rab31, RhoG, and Mcu protein expression at the *Drosophila* larval NMJ promotes aberrant synaptic phenotypes, including reductions in total bouton area and decreases in active zone puncta size. Of particular interest is the decrease in bouton active zone size in all candidate lines. Indeed, evidence suggests that the degree of BRP staining present at individual active zone sites correlates with the quantity of readily releasable vesicles and thus the probability of synaptic vesicle release [49]. Accordingly, the results obtained from this study indicate that misexpression of selected mitochondrial candidates may promote functional alterations in synaptic vesicle recycling and neurotransmission dynamics—processes that have been widely associated with age-dependent alterations in synaptic stability. Interestingly, these data are in agreement with the Hara et al. study (refer to introduction), which describes age- and mitochondrial-morphology dependent decreases in the presynaptic active zone size and the number of synaptic vesicles in NHP cortical synapses [20]. Though we report significant reductions in the size of bouton active zone puncta, quantification of the global active zone area per NMJ suggested no differences between control and candidate lines. Although speculative, this may represent a compensatory mechanism adopted by the cell to enable the maintenance of transmission and plastic properties at the synapse.

The localisation and functional specialisation of the identified candidates is wide-ranging, likely reflecting the convergence of numerous mitochondrial mechanistic pathways required for homeostatic regulation of the synaptic compartment. Rab31 is a small GTP-binding molecule localised to the mitochondrion, Golgi and plasma membrane and appears to regulate vesicular targeting, mobilization and docking [50,51]. Though there are relatively few studies documenting the constitutive role of the protein and how misexpression may modulate cellular structural and functional properties, evidence indicates that Rab31 is a member of the Rab5 superfamily of Rab GTPases [51,52]. Overexpression of the Rab5 family has been directly associated with impairments in synaptic vesicle recycling and neurotransmission at the *Drosophila* NMJ [53], although the mechanism governing this impairment remains elusive. Thus, it is probable that the age-dependent overexpression of Rab31 in synaptic mitochondria may modulate the size of the bouton active zone through a corresponding cascade, promoting concomitant synaptic dysfunction via alterations in transmission, plasticity and vesicular dynamics. Similarly, overexpression of the mitochondrial-associated protein RhoG [54] also demonstrated a reduction in bouton active zone size in addition to a highly significant reduction in total bouton area. RhoG exhibits involvement in the polymerization of the actin cytoskeleton and regulation of dendritic and axonal branching and stabilization, particularly during development [45]. Previous studies examining overexpression of RhoG report reduced axonal and dendritic complexity in vivo [55], with significant alterations in arborisation, which appear to correlate with the compacted phenotype we present here.To our knowledge, there are currently no studies describing the role of RhoG at the synaptic terminal during ageing; thus, we present novel data demonstrating that temporal increases in synaptic mitochondrial RhoG expression alter multiple morphological parameters associated with compartmental function and stability. Though the RhoG and Mcu lines exhibited quantitatively and morphologically similar phenotypes at the NMJ, divergent mechanisms likely mediate the reported synaptic alterations. Mcu is a mitochondrial calcium uniporter that regulates intracellular Ca^2+^ concentrations via the uptake of ions into the mitochondrial matrix. Due to the significant fluctuations in presynaptic Ca^2+^ concentrations during neurotransmission events, the mitochondrial Mcu has previously been associated with structural and functional perturbations in the synaptic milieu during ageing and pathogenesis [25,56,57]; however, investigations examining the effects of Mcu knockdown in vitro report conflicting results. There are reports that reductions in Mcu expression diminish vesicular mobility and release, promoting concomitant alterations in short-term synaptic plasticity [25], which suggests that regulated Mcu expression facilitates neurotransmission properties via presynaptic Ca^2+^ clearance cascades. However, studies examining synaptic vesicle recycling kinetics document no effect of Mcu knockdown on presynaptic vesicular exocytosis, total active zone area or intracellular Ca^2+^ levels [56]. Though we have not investigated the functional properties of the *Drosophila* NMJs harbouring reductions in Mcu expression, our data indicate that Mcu may modulate the size of the bouton active zone, which may affect the probability of vesicular release due to a smaller pool of readily releasable vesicles. However, whether this is due to elevations in presynaptic intracellular Ca^2+^, reductions in Ca^2+^-dependent ATP production or alterations in mitochondrial dynamics remains undetermined.

To our knowledge, this is the first study documenting the spatiotemporal alterations occurring in both synaptic and non-synaptic mitochondrial populations during advancing age. Cumulatively, our data describe that selective alterations in the aged synaptic mitochondrial proteome modulate multiple morphological parameters associated with synaptic dysfunction in vivo. Further investigations exploring how dynamic alterations in the synaptic mitochondrial proteome may attenuate neurotransmission, plasticity and vesicular mobility are required to enable elucidation of the mechanistic pathways orchestrating age-dependent impairments in synaptic function.

## Figures and Tables

**Figure 1 cells-10-03403-f001:**
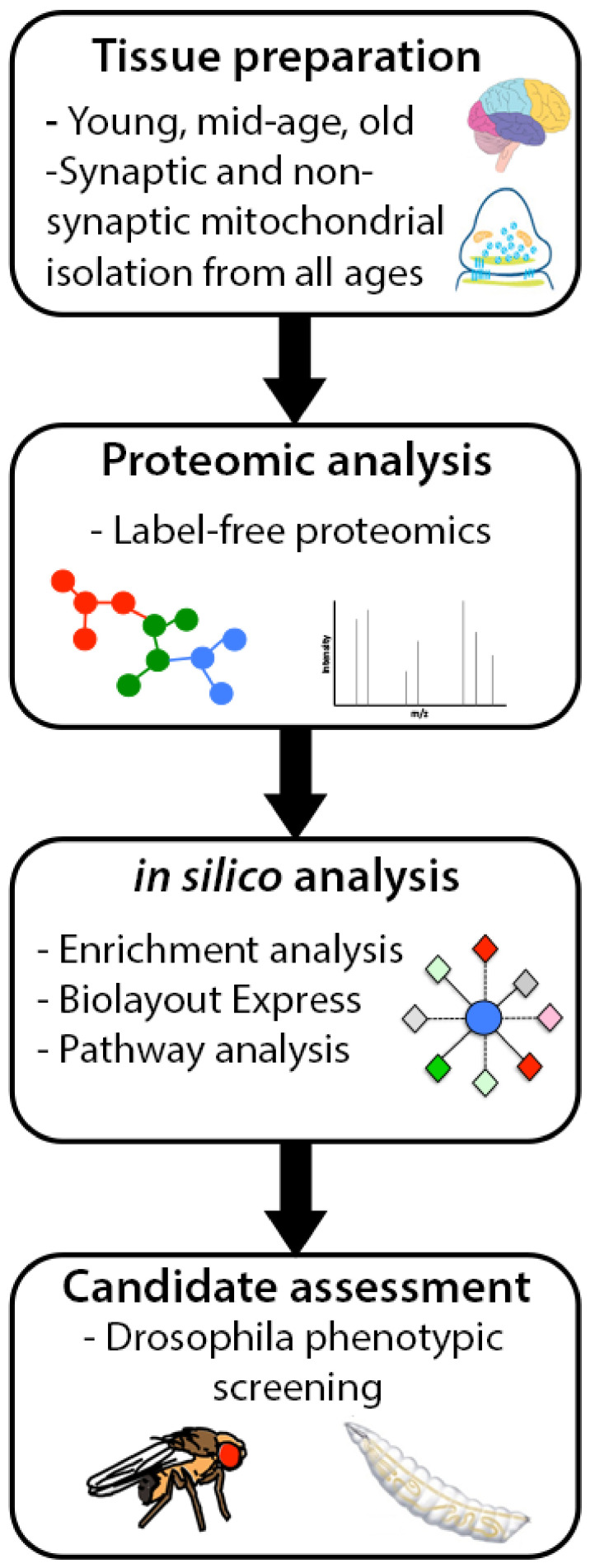
Schematic illustrating the methodological pipeline for comparison of discrete neuronal mitochondrial subpopulations throughout the ageing timecourse.

**Figure 2 cells-10-03403-f002:**
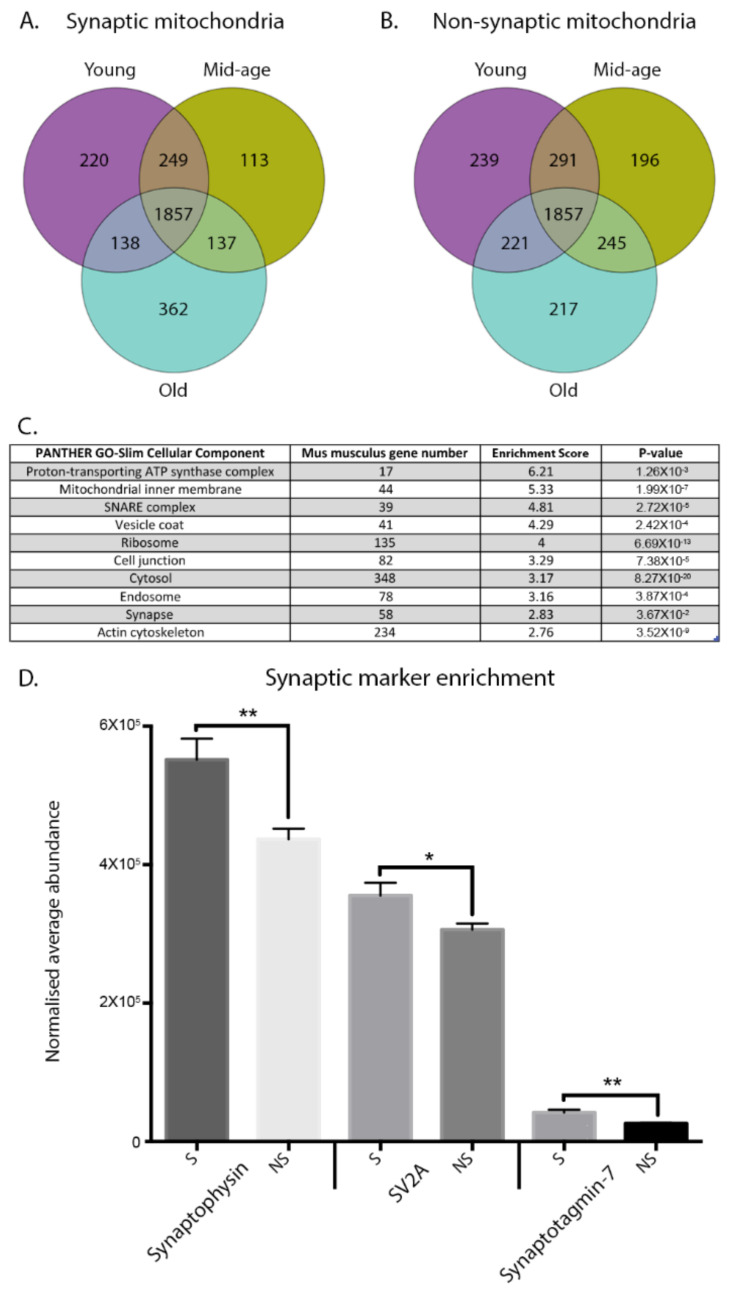
Spatiotemporal characterisation of the mitochondrial proteome. (**A**,**B**) Venn diagrams demonstrating the identification of synaptic and non-synaptic mitochondrial proteins. Proteins were identified and filtered in Progenesis software using the criteria *p* > 0.05 and 1 unique peptide. One thousand eight hundred fifty-seven common proteins were consistently identified through both the synaptic and non-synaptic mitochondrial datasets, and these were utilised for all further analyses. (**C**) PANTHER GO-Slim cellular component enrichment analysis of the synaptic and non-synaptic mitochondrial raw data. The table indicates enrichment of the mitochondrial ATP synthase complex and mitochondrial inner membrane suggesting relative purity of preparations. Fold-enrichment values greater than 1 denote overrepresentation of category in dataset. (**D**) Purity of synaptic mitochondrial preparations. The purity of synaptic mitochondrial isolates was verified with quantitative enrichment analyses utilising the raw proteomic data and the corresponding isolated non-synaptic cortical mitochondria. Comparative expression of the synaptic markers synaptophysin (*p* = 0.004), SV2A (*p* = 0.013) and synaptotagmin-7 (*p* = 0.0026) indicate synaptic enrichment of the mitochondrial preparations derived from synapses. Statistical analyses utilized unpaired two-tailed Student’s *t*-test (* *p* = 0.05; ** *p* ≤ 0.01). S = synaptic mitochondria; NS = non-synaptic mitochondria. The raw MS data and list of 1857 candidates can be accessed here: https://doi.org/10.7488/ds/3181, accessed on 15 November 2021.

**Figure 3 cells-10-03403-f003:**
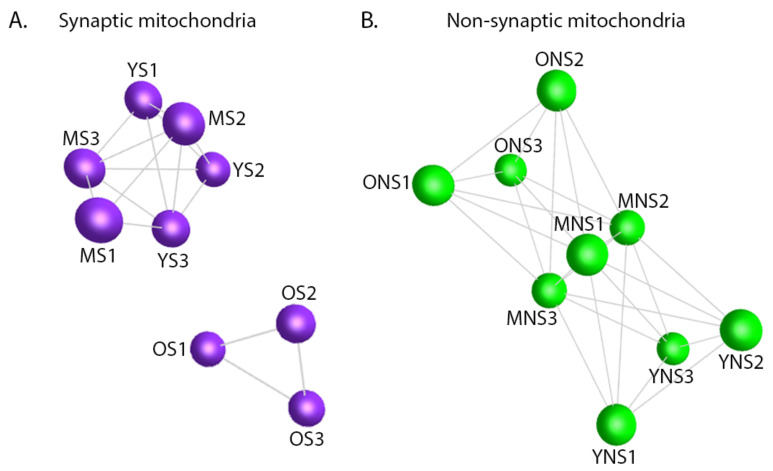
Mitochondrial ageing demonstrates subcellular heterogeneity. Synaptic and non-synaptic isolates are depicted in panel (**A**) and (**B**) respectively. Unbiased sample-sample correlation analyses generated from BioLayout Express^3D^ software. Nodes signify individual samples, and edges reflect the strength of correlation of expression. YS: young synaptic mitochondria; MS: mid-age synaptic mitochondria; OS: old synaptic mitochondria; YNS: young non-synaptic mitochondria; MNS: mid-age non-synaptic mitochondria; ONS: old non-synaptic mitochondria. All graphs clustered using Pearson’s r = 0.96.

**Figure 4 cells-10-03403-f004:**
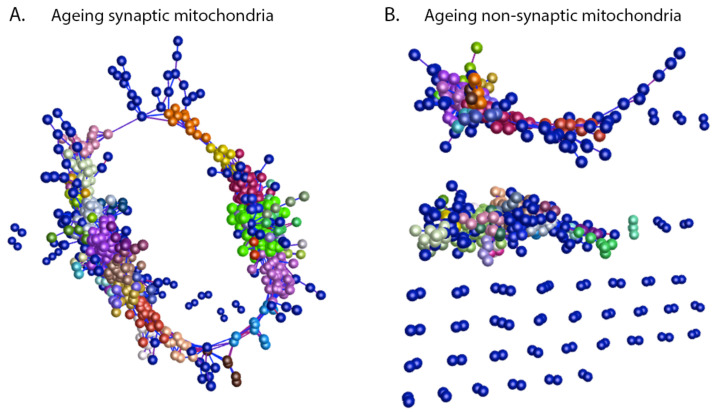
Ageing mitochondrial subpopulations (synaptic and non-synaptic depicted in panel (**A**) and (**B**) respectively) exhibit distinct protein clustering patterns. Protein-protein correlation networks displaying protein expression through the timecourse of ageing in mitochondrial subpopulations. Nodes represent individual proteins, and edges reflect the strength of correlation of expression throughout the ageing process. Thus, clusters comprised of individual nodes (i.e., proteins) are generated based on similarity in expression profile through ageing. Individual nodes are colourised based on their “membership” in clusters of proteins as defined by similarity in their expression profiles (as previously described [32]). This approach, therefore, serves to demonstrate unbiased identification of temporal trends without All graphs clustered using Pearson’s r = 0.96.

**Figure 5 cells-10-03403-f005:**
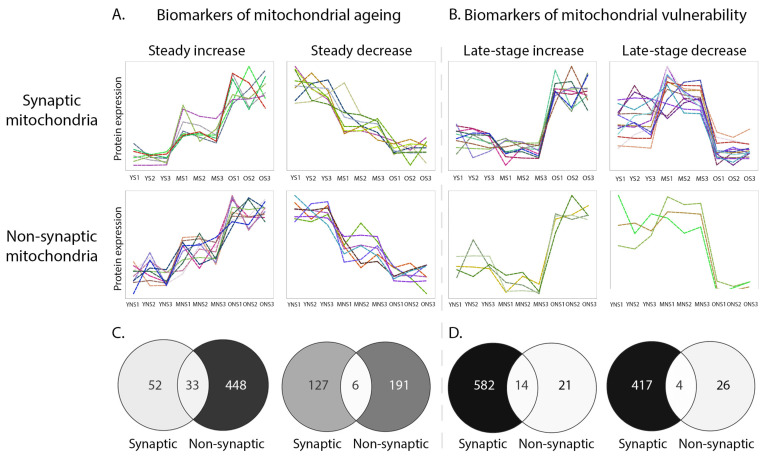
Temporal profiling of distinct mitochondrial subpopulations reveals biomarkers of organelle ageing and vulnerability. (**A**) Correlates of ageing: example temporal expression profiles of proteins demonstrating a steady up- or down-regulation during ageing. (**B**) Correlates of vulnerability: example clusters displaying late-stage increases or decreases in temporal protein expression. All graphs were generated in BioLayout Express^3D^ (r = 0.96) and display the mean protein expression across the timecourse in mitochondrial samples. (**C**,**D**) Venn diagrams indicating subtraction of candidates. Proteins demonstrating equivalent spatiotemporal profiles in both synaptic and non-synaptic mitochondria (shown at intersection) were not considered to be likely regulators of synaptic vulnerability and were therefore subtracted from further analysis. Candidates comprising panels A and B can be found here: https://doi.org/10.7488/ds/3181, accessed on 15 November 2021.

**Figure 6 cells-10-03403-f006:**
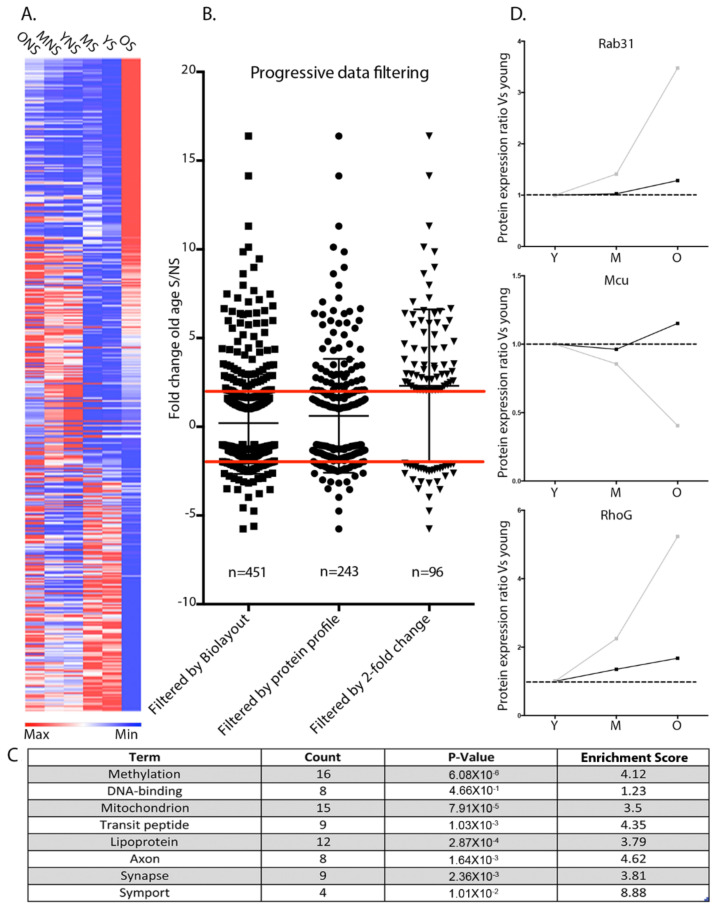
Identification of mitochondrial-associated candidates regulating synaptic stability during ageing. (**A**) Heat map displaying average normalised abundance values of 451 differentially expressed mitochondrial candidates associated with alterations in synaptic stability during ageing. Note the divergence in protein expression in the old synaptic mitochondria. Blue indicates low expression and red high expression. YS: young synaptic; MS: mid-age synaptic; OS: old synaptic; YNS: young non-synaptic; MNS: mid-age non-synaptic; ONS: old non-synaptic. (**B**) Scatterplot indicating the hierarchical filtering of proteins by stringent parameters for identification of mitochondrial candidates regulating synaptic stability. Left panel displays 451 tractable proteins identified from Biolayout Express^3D^ analyses. Middle panel represents the filtering of 451 candidates by archetypical protein expression profile. Right panel exhibits the 96 mitochondrial-associated candidates displaying archetypical protein profiles and a 2-fold change between mitochondrial subpopulations at the old age time point. Red lines indicate 2-fold change. (**C**) DAVID functional annotation analysis of 96 identified candidates that may have the propensity to modulate synaptic stability during ageing. (**D**) Candidate protein expression profiles. Graphs represent temporal expression profiles of identified candidates in both synaptic and non-synaptic mitochondria. Note the similar expression profiles through young and mid-age in both synaptic and non-synaptic mitochondria followed by significant divergence in expression at old age. All graphs display the ratio of candidate protein expression against the young age. Synaptic mitochondrial temporal protein expression: grey; non-synaptic temporal protein expression: black.

**Figure 7 cells-10-03403-f007:**
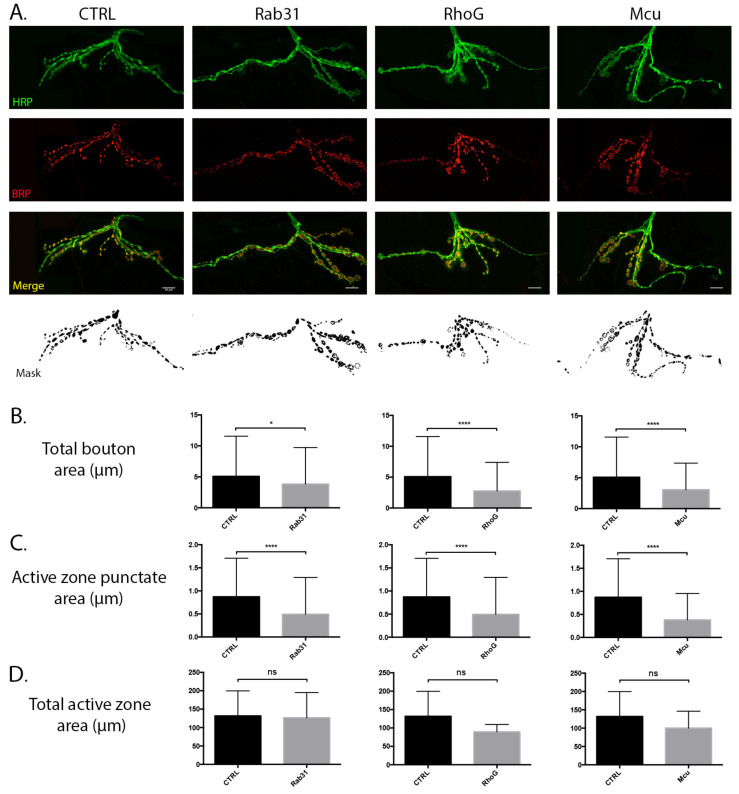
Recapitulation of candidate protein expression promotes aberrant synaptic phenotypes at the *Drosophila* NMJ. (**A**) Representative confocal images of muscle 12 NMJs, labelled with anti-HRP and anti-BRP. Overexpression of Rab31 and RhoG and RNAi-mediated knockdown of Mcu promote aberrant synaptic phenotypes at the NMJ. Lower panel displays masks utilised for quantification of morphological parameters. (**B**) Graphs represent quantification of control and candidate line total bouton area (μm). Rab31 (* *p* = 0.0376), RhoG (**** *p* < 0.0001), and Mcu (**** *p* < 0.0001) demonstrate significant reductions in total bouton area. (**C**) Graphs indicate the average size of active zone punctate (μm) in control and candidate lines. Rab31, RhoG, and Mcu all display a significant decrease in punctate size versus control (**** *p* < 0.0001). (**D**) Graphs display the total area of active zone staining in control and candidate lines. Rab31 (*p* = 0.926), RhoG (*p* = 0.3574) and Mcu (*p* = 0.5397) demonstrate no significant difference in active zone area per NMJ. All lines used the elav-Gal4 driver system. NMJs imaged at 63x. Scale bar = 10 μm, n = 5. Statistical analyses utilized unpaired two-tailed Student’s *t*-test (* *p* = 0.05; **** *p* ≤ 0.0001).

**Table 1 cells-10-03403-t001:** *Drosophila* lines utilised throughout the project. DIOPT score indicates mammalian genetic homology to ortholog lines. Higher scores denote increased candidate homology. Maximum score of 11.

Gene	Source	Annotation Symbol	*Drosophila* Stock ID	DIOPT Score
Mcu	BDSC	CG18769	42,580	9
Rab31	BDSC	CG3870	9837	2
Rhog	BDSC	CG8556	26,650	3

## Data Availability

Please use the following link to access expanded methodology and the accompanying online data repository for this manuscript here: https://doi.org/10.7488/ds/3181, accessed on 15 November 2021.

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
