# Peer review of "Temporal Profiling of the Cortical Synaptic Mitochondrial Proteome Identifies Ageing Associated Regulators of Stability"

_cells, 2021, doi:10.3390/cells10123403_

Round 1

Reviewer 1 Report

Authors studied changes of the proteome in synaptic and nonsynaptic mitochondria caused by ageing. They identified differentially expressed proteins in discrete timepoints on synaptic and nonsynaptic mitochondrial fractions and these differences were used to select proteins potentially involved in processes leading to vulnerability of synaptic mitochondria. Three proteins (Rab31, RhoG 444, and Mcu) were selected for verification in Drosophila larvae, their manipulated expression showed aberrant synaptic phenotypes.

The authors used appropriate analytical methods and conclusions are supported by their results.

Reviewer 2 Report

Synapses are particularly susceptible to the effects of advancing age and mitochondria have long been implicated as organelles contributing to this compartmental vulnerability. How the mitochondrial molecular cascades promoting age-dependent synaptic demise remain to be elucidated.

Therefore, the authors of this study set the objective to investigate how the synaptic mitochondrial proteome was dynamically and temporally regulated throughout ageing to determine whether protein expression changes within the mitochondrial milieu are actively regulating the age-dependent vulnerability of the synaptic compartment.

The proteomic analysis of wild-type mouse cortical synaptic and non-synaptic mitochondria across the lifespan revealed significant age-dependent heterogeneity between mitochondrial subpopulations, with aged organelles exhibiting unique protein expression profiles.

Recapitulation of aged synaptic mitochondrial protein expression using molecular genetic tools in vivo revealed several novel functional mediators that have the propensity to significantly modulate multiple morphological parameters at the synapse, suggesting that the mitochondrial proteome and synaptic morphometry are intimately intertwined. The data indicate that selective alterations in synaptic mitochondrial protein expression may, in part, mediate enhanced vulnerability of the cortical synapse during advancing age.

Positive aspects to highlight:

  • The authors do not bred animals specifically for this project.
  • The authors consider gender when designing the experiment.
  • The materials and methods are written in detail, facilitating the reproducibility of the work. The mitochondrial isolation protocol is very laborious and meticulous.
  • In the label-free proteomics study, appropriate parameters as well as astringent have been used to ensure the reliability of the results.
  • The authors examine whether alterations in candidate protein expression in aged synaptic mitochondria was contributing to the vulnerability status of synaptic compartments in vivo! Which provides a physiological dimension of the results.
  • The discussion is extensive and adequately defends the results obtained.

Therefore:

Overall recommendation: Accept after Minor Revision.

REVISION:

  1. Methods

1.1 Line 147._ It is advisable to put the pH at the end of the buffer composition.

1.2 Line 151._ “initially for 4 h…” at what temperature?

  1. Results

2.1 Fig. 2A&B._ The number of common proteins in synaptic mitochondria at different ages (1857) is the same as for non-synaptic mitochondria. It’s a coincidence?

2.2 Line 235._ Reference should be made to figure 2C.

2.3 Fig. 2D._ How do the authors explain that synaptic markers are so abundant in non-synaptic mitochondria?

2.4 Line 328._ Can the authors explain in more detail how have calculated that they obtain 451 proteins?

2.5 Line 354._ I think the authors refer to Figure 6A.

2.6 Line 357._ Reference should be made to figure 6C.

2.7 Line 361._ Why did the authors choose the RhoG candidate for the regulatory role study?

Reviewer 3 Report

Synaptic neurotransmission is a highly energy demanding process, which is tightly regulated by Ca2+ signaling. In this context, mitochondria play a pivotal role in ATP production and calcium buffering in order to power fluent synaptic function. In recent years, several studies implicated that a decline in mitochondrial function is associated with impaired synaptic activity. Changes in the mitochondrial proteome even might serve as driver of biological aging and as cause of age-related diseases. Yet, the specific alterations in mitochondrial proteomes caused by aging are largely unknown, particularly in specific tissues such as neurons.

In this study, Graham et al. tested in vivo the dynamic change of the mitochondrial proteome during different stages of ageing in mice. Their aim was to connect these temporal changes with the age-dependent vulnerability of the synapse. With the help of proteomics and a subsequent in silico analysis of their data the authors intended to analyze mitochondrial age-related perturbations. Unfortunately, this analysis is very superficial and does not provide insights into which proteins actually change during aging. As it stands, this study simply confirms that the protein content of mitochondria in synapses of old mice is somewhat different to that in young animals. Nevertheless, the overall topic is interesting and with a considerable revision, this study might become appropriate for Cells.

Major points:

  1. It is essential to test the functional state of the mitochondrial fractions that were compared here. A Seahorse experiment is the standard in the field. This is an essential prerequisite before any conclusions can be drawn on the (respiratory) activity of the mitochondria isolated from the 12 animals used here.
  2. It is unclear why 12 animals for used for three times three samples. This needs to be clarified. Are these three replicates indeed three independent biological replicates? This would be essential.
  3. 3A suggests that the old synaptic mitochondria differ from young and middle aged ones. This is the main conclusion from this study. The authors should show volcano plots to compare the proteomes (abundance and significance) of OS1-3 with MS1-3. In these volcano plots they should color-label proteins found in the MitoCarta set and indicate those mitochondrial proteins that are particularly increased or reduced in old synapses. The names of these significantly changed proteins should be shown in tables either in the main manuscript or as supplements.
  4. All proteomics data need to be made available to the authors, e.g. as Excel files.
  5. In its current form, Fig. 4 carries no relevant information. Either it should be removed or the authors make the effort to define specific mitochondrial activities/complexes/GO terms that are affected by aging.
  6. The definition of age-affected proteins shown in Fig. 5A is potentially interesting. Again, the authors have to provide names to these proteins so that readers can use this information. As it stands, the figure just shows that some proteins change with age which is a no-brainer.
  7. 6C makes the point that 15 mitochondrial protein significantly change with age. The names and functions of these proteins should be listed and discussed.
  8. It is unclear why authors chose Rab31 and RhoG as candidates for their follow-up experiments as both are no mitochondrial proteins but cytosolic G proteins.
  9. The intention of a proof-of-principle experiment in Drosophila is honorable, however, as it stands the value of Fig. 7 is not clear. Either the authors clearly improve this analysis or, even better, remove it from this study and show more compelling data in a separate study in the future.

Specific points:

  1. In Figures 2C and 6C the authors show a GO term analysis as a table. They should better show this as a figure. The p-values should be below 0.05 and only significant changes should be shown. The authors might show the components on the y-axis and the enrichment score on the x-axis. The p-value can be shown by different colors according to significance.
  2. In chapter 3.3, a conclusion summarizing the message from the results should be added.
  3. In the text, the reference for Figure 6A is missing and should be included.
  4. It is not entirely clear, why the candidate testing was done on the one hand in Drosophila and on the other hand in larvae. This should be clearly stated in the text, or as discussed before, the fly experiments should be removed.
  5. To Figure 6A the authors should add names to the heatmap, at least for the candidates discussed later in the text. The same is also true for figure 6B. It would be helpful for the reader if the proteins are colored in the plot.
  6. In chapter 3.4 the authors state that ‘Manipulation of Rab31 (9837), RhoG (26650) and Mcu (42580) expression revealed striking synaptic phenotypes at the third instar larva muscle 12/13 NMJ in multiple morphological parameters associated with the structural and functional stability of the synapse’. Here a further description of what morphological parameters are changed would improve the understanding of the section.

Round 2

Reviewer 3 Report

The authors commented on the points raised on the previous submission. In its present form, the study might be of interest for some specialized readers.